# MAC protocol with grouping awareness GMAC for large scale Internet-of-Things network



Abdulrahman Sameer Sadeq[1], Rosilah Hassan[1], Azana Hafizah Mohd Aman[1], Hasimi Sallehudin[1], Khalid Allehaibi[2], Nasser Albogami[2] and Anton Satria Prabuwono[3]

[1] Center for Cyber Security, Universiti Kebangsaan Malaysia UKM, Bangi, Selangor, Malaysia
[2] Department of Computer Science, King Abdulaziz University, Jeddah, Saudi Arabia
[3] Department of Information Technology, Faculty of Computing and Information Technology in Rabigh, King Abdulaziz University, Jeddah, Saudi Arabia

## ABSTRACT

The development of Medium Access Control (MAC) protocols for Internet of Things should consider various aspects such as energy saving, scalability for a wide number of nodes, and grouping awareness. Although numerous protocols consider these aspects in the limited view of handling the medium access, the proposed Grouping MAC (GMAC) exploits prior knowledge of geographic node distribution in the environment and their priority levels. Such awareness enables GMAC to significantly reduce the number of collisions and prolong the network lifetime. GMAC is developed on the basis of five cycles that manage data transmission between sensors and cluster head and between cluster head and sink. These two stages of communication increase the efficiency of energy consumption for transmitting packets. In addition, GMAC contains slot decomposition and assignment based on node priority, and, therefore, is a grouping-aware protocol. Compared with standard benchmarks IEEE 802.15.4 and industrial automation standard 100.11a and user-defined grouping, GMAC protocols generate a Packet Delivery Ratio (PDR) higher than 90%, whereas the PDR of benchmark is as low as 75% in some scenarios and 30% in others. In addition, the GMAC accomplishes lower end-to-end (e2e) delay than the least e2e delay of IEEE with a difference of 3 s. Regarding energy consumption, the consumed energy is 28.1 W/h for GMAC-IEEE Energy Aware (EA) and GMAC-IEEE, which is less than that for IEEE 802.15.4 (578 W/h) in certain scenarios.

# INTRODUCTION

The recent development of Wireless Sensor Networks (WSNs) and the incorporation of technologies of Internet of Things (IoT) has enabled their applications in various industrial fields, particularly through IoT-based WSN (IoT-WSN) (*Hassan et al., 2020*). Such the emergence has led to numerous applications in different sectors such as agriculture (*Hassan et al., 2020*; *Keswani et al., 2018*), smart cities (*Al-Majhad et al., 2018*; *Nassar et al., 2019*; *Zhang, 2020*), intelligent transportation system (*Muthuramalingam et al., 2019*),

Corresponding author
Rosilah Hassan,
rosilah@ukm.edu.my

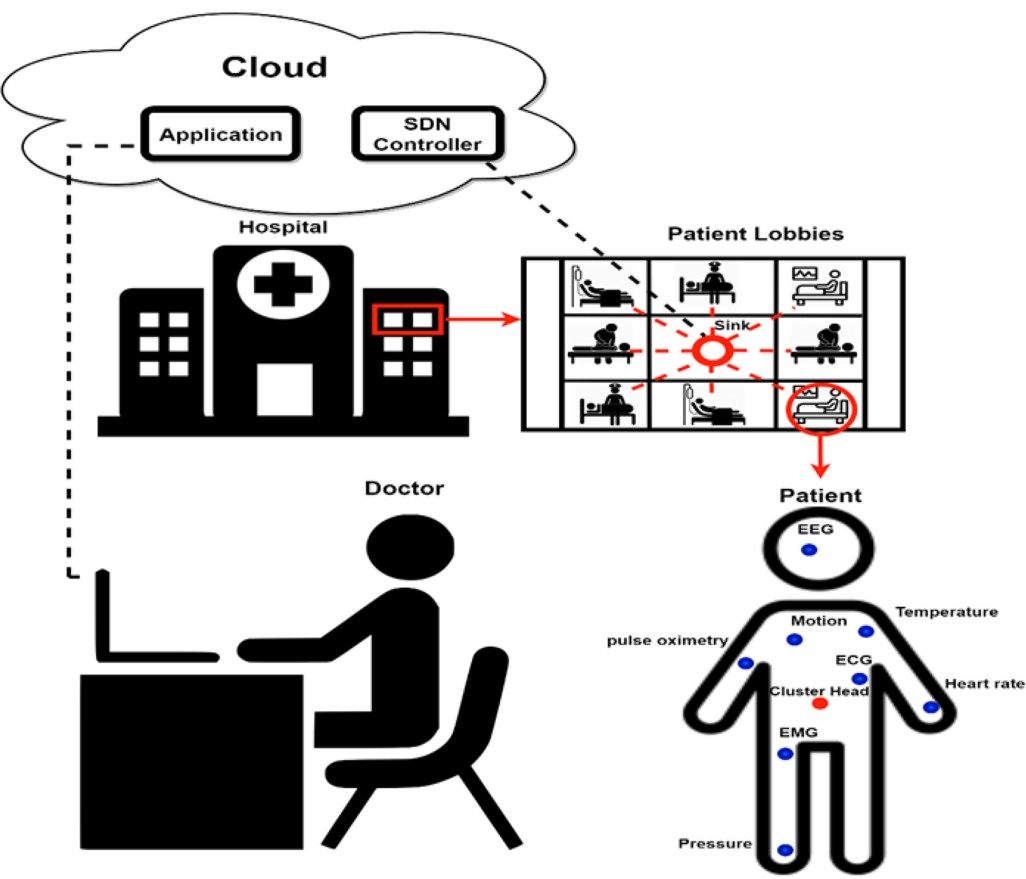

**Figure 1 Application of IoT-WSN in health care using SDN technology.**

medical field (*Onasanya & Elshakankiri, 2019*; *Yao et al., 2019*), security and surveillance (*Benzerbadj et al., 2018*; *Memon et al., 2020*), military (*Zieliski, Chudzikiewicz & Furtak, 2019*), forensics (*Yaqoob et al., 2019*), education, and voting (*Srikrishnaswetha, Kumar & Rashid Mahmood, 2019*). Sensing-based applications that monitor and gather data are regarded as common applications of IoT (*Sadeq, Hassan & Mahdi, 2018*; *Wu, Wu & Yuce, 2018*). Figure 1 shows a conceptual diagram of a healthcare monitoring application using IoT-WSN. Body area networks are installed on patients in hospitals, and they continuously gather data from all patients in real time. The sensors are deployed in 3D based configuration, and the sensors are located in each patients' room separately from other patients' rooms, which emphasize the assumption of clusters-based decomposition. The collected data are then used within an intelligent system to assign care algorithmically to increases the recovery ratio. For one floor, one sink connects the clusters, each representing one patient with sensors on different parts of the body. On the other end, the sink is connected to a Software Defined Network (SDN) controller connected to an application layer in the cloud to monitor patients and assign tasks to doctors.

This application requires continuous sensor data collection and transfer to the cloud. The wireless nature of the network and the limited resources of its nodes create two issues.

First, the management of node access to the medium must be coordinated with consideration of the sensing rate, sensors' nature, and their relation to the application. This issue affects Quality of Service (QoS) metrics in the network. Second, the management of energy in the network affects the lifetime metric. These issues are not independent of each other, enabling Carrier Sense Multiple Access/Collision Avoidance (CSMA/CA) mode may cause failure in several sensors in accessing the medium, which eventually leads to energy waste and shorter lifetime. However, saving the node energy requires careful scheduling and management of their access to the medium.

Numerous types of Medium Access Control (MAC) protocols are available. A few of the most widely used are IEEE 802.15.4 and ISA 100.11a as IoT MAC protocols in numerous types of applications. These types have small differences in terms of enabling or ignoring packet life time or adding priority to packets or not. One critical issue of these two protocols is the scalable energy awareness solution (*Sotenga, Djouani & Kurien, 2020*). When an application requires installing a high number of sensor nodes, the protocol may be inefficient in terms of energy-saving due to the resulting collisions. Another issue is the non-awareness of various priority and grouping aspects of the sensor nodes. Such an awareness is important to manage the medium effectively. Recent variants are developed with grouping awareness, such as User-Defined Grouping (UDG) (*Yasari et al., 2017*) based on ISA 100.11a. However, this variant has various limitations and necessitates developing of IoT-oriented MAC protocol with scalability, energy efficiency, and grouping awareness. The article aims to propose a novel protocol based on both IEEE 802.15.4 and ISA 100.11a to improve their CAP performance based on exploiting prior knowledge about the clustering information of the network and the priority level of the nodes. Therefore, this study aims to develop two novel variants of Grouping MAC (GMAC) based on current benchmarks, namely, GMAC-IEEE and GMAC-ISA.

## LITERATURE REVIEW

In addition, several improvements on IEEE 802.15.4 in various aspects have been performed, such as the Clear Channel Assessment (CCA) and its effect on the delay and overhead on the protocol. An improvement on CCA (*Wang, Liu & Yin, 2018*) is proposed using a graded tailoring strategy, which checks the length of the original packet and modifies its original size according to the partition points. Assuming that the same back-off unit size is used, the protocol includes 20 symbols. To make the data packet tail size eight, those with lower than eight add zeroes, and those higher than eight subtract zeroes. This improvement is useful from the general perspective of delay and over-head, but it ignores prior knowledge of nodes or packets priorities. The mechanism of CCA of IEEE 802.15.4 is also examined in different ways. For example, the CCA has been modified to include primary and secondary stages (*Gamal et al., 2020*). In addition, an optimization model is built for the delay with energy consumption as a constraint. The model is solved using linear quadratic programming, but it does not consider retransmission essential in IEEE 802.15.4.

Another modification to IEEE 802.15.4 (*Patel & Kumar, 2017*) aims to increase the number of CCA from one to two, reducing the number of back-off periods to confirm the

status decision of the channel and scarify the low energy consumption of CCA. This modification avoids high-energy consumption when a failure occurs and bandwidth loss if the channel becomes idle. The number of retransmissions and their effect on performance is also examined. The network nodes are divided into sub-groups or classes according to the number of failed retransmission (*Henna & Sarwar, 2018*). Specifically, the low number of failed retransmissions implies low increases in the back-off time and converts the protocol of IEEE 802.15.4 from a fixed to an adaptive back-off. However, the approach lacks an automatic means to decide to change the back-off for each class. Another issue is the neglect of energy level of each sensor that is considered a highly critical aspect in the performance. Furthermore, the approach does not embed prior knowledge regarding the sensor's class or priority related to its function in the system.

TDMA and CSMA functionalities of IEEE 802.15.4 are also combined for WSN scheduling with the support of demand, that is, profile. A proposed WSN scheduling based on the concept of network virtualization (*Uchiteleva, Shami & Refaey, 2017*) divides the networks into profiles, each of which indicates a set of nodes sharing the same channel demand nature or characteristics. The scheduling proposes two profile categories, bursty and periodic. The super-frame in IEEE 802.15.4 is then decomposed into contention access frame and contention free frame (*Uchiteleva, Shami & Refaey, 2017*), which contains a set of guaranteed time slot. Next, an optimization is conducted to maximize the utility for each profile. The algorithm uses a greedy optimization approach. Furthermore, as stated in (*Shrestha, Hossain & Choi, 2014*), the strength of CSMA/CA when it is combined with TDMA improves the scalability by preserving the performance of legacy-based CSMA/CA-based MAC scheme in congested networks.

Other approaches in the literature aim to improve ISA 100.11a, which is regarded as a common protocol for industrial wireless sensor networks owing to its wide use in MAC layer management of sensors of control systems (*Florencio, Doria Neto & Martins, 2020*). In a recent survey (*Raptis, Passarella & Conti, 2020*), a comparison between ISA 100.11a and WirelessHART has been conducted to conclude the need to optimize various aspects in ISA such as communication and energy optimization. An optimization of ISA under TDMA (*Satrya & Shin, 2020*) proposes a solution representation that provides a code for each node according to its time slot. Next, the work of (*Yasari et al., 2017*) develops a genetic-based scheduling algorithm that enables flexible scheduling in ISA 100.11a. However, this work only optimizes one parameter in ISA 100.11a, the packet lifetime assigned to each group, and another parameter outside ISA 100.11a, which is the number of nodes in each group. An objective function is then used to maximize the number of nodes and the distribution of nodes in the groups according to their weights. In addition, this work ignores a direct optimization to the network performance measures such as QoS, which is used as constraint in the optimization only. Some researchers aimed to enhance ISA 100.11a in the context of application, such as adapting ISA to operate in a specific control environment. In (*Herrmann & Messier, 2018*), an optimization of the scheduling and the routing (cross-layer) is proposed. The goal is to minimize energy consumption and prolong the lifetime of the petroleum refinery process. The frame structure and an optimization of the scheduling and selection of the

routing hops, are the elements of ISA enhancement. Despite the many developments of ISA 100.11a and IEEE 802.15.4 and other related WSN scheduling, some researchers proposed different improvement perspectives. For example, in (*Farayev et al., 2020*), the pre-knowledge of the periodic nature of data generated in the network is exploited to formulate joint optimization of scheduling, power control, and rate adaptation for discrete rate transmission mode. Although this assumption is useful when it is valid, many WSNs have no pre-knowledge of the nature of data generation, such as event-based monitoring.

The literature on MAC layer scheduling in IEEE 802.15.4 and ISA 100.11a tackles various aspects of these two protocols. Several approaches focus on optimizing CSMA (*Gamal et al., 2020*) and others pay attention to TDMA (*Osamy, El-Sawy & Khedr, 2019*), but limited work concentrates on integrating both functionalities (*Wang, Liu & Yin, 2018*). Furthermore, none of the previous work develops protocols for integrated CSMATDMA with grouping awareness. This factor affects the performance that relies heavily on prior knowledge about each node in terms of its application or role in the system or its priority of responding when packets are generated, compared with systems that consider scheduling of medium access but not the source or group of node priority.

Overall, the handling of the problem of MAC scheduling in IEEE 802.15.4 based protocols, despite its covering to various development aspects such as optimization of parameters, incorporation of adaptive approaches, usage of TDMA and/or CMSA, non-of the previous approaches has addressed the scheduling with a consideration of the node's geographical distributions. Considering that the collisions occur more frequently when the nodes are close to each other or share the same coverage zone, relative location-aware contention is an important criterion for optimizing the scheduling and reducing the collisions. The article aims to propose a novel MAC scheduling protocol based on IEEE 802.15.4 that enables clustering, which is location-based grouping as a criterion for handling scheduling. Furthermore, the proposed protocol will exploit TDMA to exchange clusters' information with the sink and CMSA for transmitting information within the clusters to the cluster heads. With such scheduling management, the developed protocol is the first MAC scheduling that jointly enables clustering awareness and CSMA-TDMA integration in a single protocol.

## MATERIALS & METHODS

This section provides the developed protocol and the evaluation methods and metrics used to compare with state-of-the-art protocols or benchmarks. We divide the methodology into several parts. At first, we present the assumptions and symbols, and the network hierarchy. Next, the energy model and the energy-aware back-off time along with the protocol design are provided. In the protocol design, we provide the network cycle and the protocol activity diagram.

### Assumptions and symbols

We assume that the network is represented by a graph $G = (V, E)$ with $V = \{n_1, n_2, \ldots n_K\}$, where $n_i$ denotes the sensor $i$, and k denotes the number of sensors.

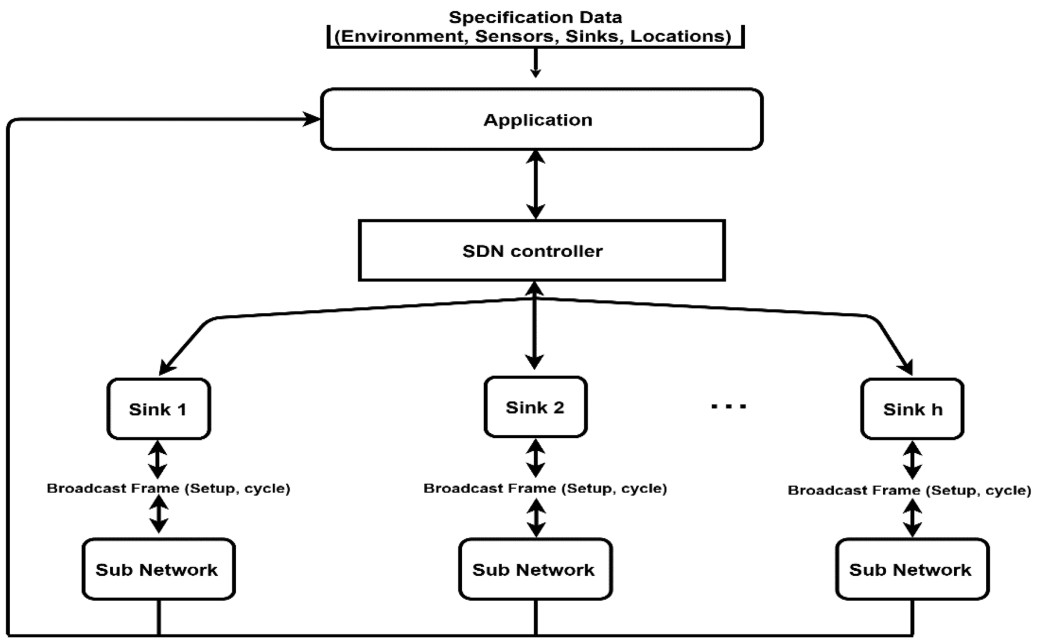

**Figure 2 Conceptual diagram of the network hierarchy.**

Each sensor $i$ is located in position $(x_i, y_i, z_i) \in R^3$. $E$ denotes the set of links between the nodes.

The clustering information, the number of groups in the network, and the priorities of the nodes are defined in advance based on the application.

The network consists of non-overlapping clusters in the coverage of sensor nodes when using low power transmission mode, but cluster heads connect to the sink when using high-power transmission mode.

The network consists of several priority groups $L_j = 1, 2..maxGroup$, where $maxGroup$ denotes the maximum number of groups of sensors, and $j$ denotes an index of the priority level. Lower j is equivalent to higher priority according to Eq. (6).

Each node creates an array with size equal to the number of slots. The value in the array indicates the probability of selecting one of the slots for transmission. Initially, all the slots are assigned the same value, which means that no slot has higher probability than another.

Each sensor node is equipped with a battery, and the initial energy for all sensor nodes is the same.

## Network hierarchy

We build a GMAC protocol based on a network hierarchy depicted in Fig. 2. The information on environment, sensors, locations, and sinks are provided to the application responsible for managing the network. The information is analyzed, and the optimal cluster decomposition and cluster head assignment are generated and provided to the sinks by the SDN controller through the flow table. Each sink knows its clusters, and their cluster heads become responsible for collecting the data to send to the cloud. GMAC protocol

operates within this sub-network that is assumed to be non-overlapping in the coverage zone when using low transmission mode, which is only enabled within cluster communication. In the beginning, the sink broadcasts an information frame that carries the cycle specification, slots, and their definitions. The details of this protocol are presented in the following sub-section.

## Energy model

Energy is consumed at each sensor whenever a data packet is sent or received. The consumed energy is calculated according to number of bits in the packet for transmission and receiving and the number of bits in the packet and the distance between the sender and receiver in the transmission case. We also assume that the sensors can operate in one of two modes. The first is the high energy mode for communicating between cluster heads and sinks, and the second is the low energy mode for communicating within clusters. This model is based on the radio energy dissipation model presented in Eqs. (1) and (2) that are given in (*Wang et al., 2017*).

$$E_{Tx}(k, d) = E_{Tx-elec}(k) + T_{Tx-amp}(k, d)$$

$$= \begin{cases} E_{elec}*k + \varepsilon_{fs}*k*d^2, & d \leq d_0 \text{ for low energy mode} \\ E_{elec}*k + \varepsilon_{mp}*k*d^4, & d > d_0 \text{ for high energy mode} \end{cases} \quad (1)$$

To receive k bits' message, the energy consumption is given in Eq. (2).

$$E_{RX}(k) = E_{RX-elec}(k) = E_{elec} * k \quad (2)$$

## Energy-aware back-off time

One of the developed models of the protocol is the energy-aware back-off time which enables the node to consider its residuals energy. The approach uses the current energy in the node $E$, the minimum allowed energy $E_{min}$, the maximum energy $E_{max}$, the maximum and minimum values of $CW$ or $CW_{max}, CW_{min}$ in a linear proportional model as it to select the best $CWE$ as it is given in Eq. (3) which will be used in the back-off time calculation in Eq. (4).

$$CWE = \left(\frac{CW_{max} - CW_{min}}{E_{max} - E_{min}}\right)(E - E_{min}) + CW_{min} \quad (3)$$

$$Back \ off \ Time = Random(0, CWE) \times Slot \ Time \quad (4)$$

Considering that at certain point of time, the nodes will have different values of residual energies, then it is more likely when two nodes contribute in a collision when attempting to access the channel, they have different energies. Hence, the Packet Delivery Ratio (PDR) will get more chance to increase, and node will be more immune from wasting its energy in frequent collision. The suffix EA will be used to indicate to the protocol that uses this developed model.

## Protocol design

GMAC is a MAC scheduling protocol for WSN that is partitioned into multi-clusters and six cycles. Each cycle contains one frame, except cycle 1 that includes $N_C$ frames. Hence, the total number of frames is $5 + N_C$. Next, we present the details of the cycles, the general protocol activity diagram and their corresponding frames.

1) Cycle 0. One-time cycle that occurs only at the beginning of the network. In this cycle, the frame is sent from the sink as a broadcast frame to identify the cluster heads, nodes, and the assignments. The frame is a broadcast frame, and the nodes are in low power mode. In case of packet loss within this cycle, the same settings of the previous cycles are used.

2) Cycle 1. A periodic cycle to send data from the nodes to the cluster head. The nodes operate in low coverage mode, and the frames in the cycle differ from one cluster to another according to cluster size and the group information. The frame of any cluster $i$ contains the number of slots according to Eqs. (5) and (6). Each subframe $i$ has the size of

$$Nf_i = m \sum_{j=1}^{L} L_j N_j^i \tag{5}$$

$$L_j = maxGroup - j + 1 \tag{6}$$

where, *maxGroup* denotes the maximum number of groups in the sensors, and j denotes an index of the group. Low index is equivalent to high priority level given to the group.

$N_j^i$ denotes the number of nodes in cluster $i$ that have the priority level $L_j$. The number of slots that are assigned to priority level $L_j$ in sub-frame equals to $L_j N_j^i$ and denoted by $xg_j^i$. Thus,

$$Nf_i = \sum_{j=1}^{L} xg_j^i \times m$$

The nodes inside group $j$ has the right to compete for its slot only inside the sub-frame $i$, avoiding wasted energy of the nodes competing with those of different groups.

Example 1: Assume a cluster i with combined four groups and five nodes in each group. The frame structure is

$$Nf_j = 5(1 \times m + 2 \times m + 3 \times m + 4 \times m) = 5(1 + 2 + 3 + 4) = 50,$$
$$\text{where m} = 1. \tag{7}$$

The equation of $N_f$ can be generalized to consider the number of nodes $N_i$ in group i when the number of nodes differ in each group, such as

$$Nf_j = \sum_{j=1}^{L} L_j m \times N_j^i \tag{8}$$

3) Cycle 2. A periodic cycle to send data from cluster heads to sinks. In this cycle, the cluster heads operate in high coverage mode while other nodes are in sleep mode. As provided in Eq. (9), the frame type of this cycle is long and setup by the coordinator based on the cluster sizes and number. On the one hand, each cluster head has a predefined number of slots according to its cluster size. In this frame, cluster heads do not compete. On the other hand, the nodes in each cluster are in sleep mode while their cluster head is communicating with the sink, to prevent interference. The frame is decomposed of $N_c$ subframes where each is assigned to one cluster head. After the cluster head selection by the controller, each cluster head knows its dedicated subframes for transmission. The sub-frame sizes are determined on the basis of the number of nodes in their corresponding cluster and their groups. The frame consists of the number of slots equal to $N_{tf}$.

$$N_{tf} = m \sum_{j=1}^{Nc} \times \sum_{i=1}^{Lj} L_j \times N_i = \sum_{j=1}^{Nc} f_j \qquad (9)$$

Example 2: Assume a network of four clusters. Table 1 shows that each cluster contains different numbers of groups and nodes inside each group. Apply Eq. (7) to calculate the total size of the sub-frame for the corresponding cluster when m = 1. To explain the sequence or the network cycle in this example, we show the detail frame and sub-frames in Fig. 3.

4) Cycle 3. A periodic cycle triggered every $T_u$ to update the energy status and packet delivery status of the nodes to their cluster heads. The cycle repeats after the NC number of cycles that are predefined by the application. Nodes work in low coverage mode. Only one frame, update frame 1, is contained, and its length is the same as in cycle 1.

5) Cycle 4. A periodic cycle triggered every $T_u$ to update the energy and PDR status of the nodes to the sink. In this cycle, a long frame is sent by cluster heads to the sink that has predefined slots. The cluster heads operate in high coverage mode while other nodes are in sleep mode. The frame type, update frame 2, is a long frame as provided in Eq. (9).

6) Cycle 5. A periodic cycle triggered every $T_u$ and broadcasted by the sink to change the cluster heads based on the node energy consumption. Only one frame is contained, update frame 3, to identify the new head of a certain cluster. In addition, the cycle contains an indicator variable that defines one of three possible states: State 1 allows contention only for nodes that have not occupied their slot, State 2 allows contention for all nodes, and State 3 does not allow any contention.

7) Network Cycle. Figure 3 shows that the network consists of six cycles. In cycles 0, 1, 4, and 5, the nodes work in low power mode wherein the coverage radius is regarded as RA m. This value assures no interference between nodes in one cluster with those in another cluster. In cycles 2 and 3, the cluster head works in high power mode in which the radius becomes RB and is responsible for sending data from the cluster head to the sink. Nodes in high power mode go on sleep mode except for the cluster head.

8) Protocol Activity Diagram. Figure 4 shows the activity diagram that provides sequential descriptions of the various cycles and tasks in the GMAC protocol. The class diagram starts with having a broadcast frame to inform all the nodes about clustering and the cluster head decision. This application layer makes this decision in the cloud based on various considerations, which is beyond the scope of this study. This broadcast frame includes information of each cluster of each node, its cluster head, and its dedicated slot in the TDMA cycles. The sizes of those cycles and contention access periods and their sizes are also defined.

Next, each cluster's nodes repeatedly send data to its cluster head using CAP period of ISA or IEEE frame. Subsequently, TDMA cycle is performed to send the data gathered by the cluster head to the sink in sequential manner. We define a time period $T_u$ that updates the cluster head and the sink on node energy status. This awareness enables maximum utilization of slots by updating the indicator variable that allows the nodes to increase this contention, and consequently, maximize its rewards. Hence, three consecutive cycles are triggered every $T_u$: cycle 3 for having nodes updating its cluster head using TDMA about its energy and nodes occupation status, cycle 4 for having the cluster head update the sink about their cluster states using TDMA, and cycle 5 when the sink updates the entire network about the clustering decision and the indicator variable. The pseudocode of the whole protocol is presented in Algorithm 1.

## EXPERIMENTAL DESIGN AND RESULTS

This section provides the evaluation scenarios and experimental works for comparing our GMAC with the benchmarks IEEE 802.15.4, ISA 100.11a, and UDG (*Yasari et al., 2017*). GMAC has two variants, namely, GMAC-IEEE and GMAC-ISA. Hence, we have five protocols: GMAC-IEEE, GMAC-ISA, and UDG based on ISA 100.11a, IEEE 802.15.4, and ISA 100.11a.

### Experiment design

The simulation work uses MATLAB 2019b for evaluation. The experiment design is based on changing the inter-arrival time that indicates the offered load in the network. The range of the inter-arrival time changes from 0.1 to 5 s. In each experiment, a set of scenarios are generated on the basis of changing two variables, the number of nodes in the network and the number of clusters. The number of nodes is $K = 10, 20, …100$, and the number of clusters is taken as $N_c = 1, 2, ..10$. In addition, different priority groups are generated $L_j = 1, 2, ..10$. Moreover, the average number inside each cluster ranges from $\frac{K}{N_c} - \sigma$ to $\frac{K}{N_c} + \sigma$ where $\sigma$ denotes the diversity of the number of nodes in the clusters. Table 2 provides the simulation parameter. For channel fading, we use Nakagami, which is a method commonly used in the simulation of physical fading radio channels. Using the parameter, this distribution can model signal fading conditions ranging from extreme to mild, light to no fading, (*Beaulieu & Cheng, 2005*). The scenario details in terms of number of nodes and number of clusters are shown in Table 3 with the visualization of two scenarios in Figs. 5A and 5B.

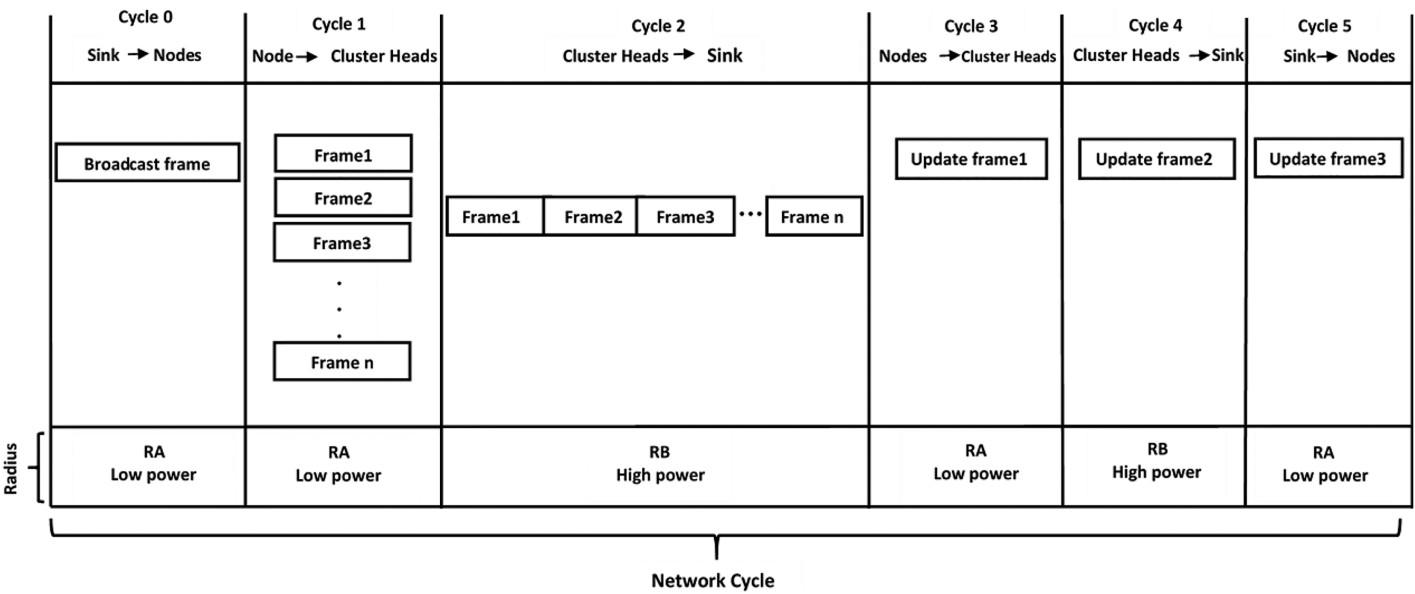

**Figure 3 Total network cycles with the corresponding frame.**

**Table 1 Example values for group priority and nodes numbers.**

| Cluster number | Group$_{priorityLevel}$ | $L_j$ | Nodes inside group | Frame size |
|---|---|---|---|---|
| C1 | G1 | 6 | 3 | 58 |
|    | G2 | 5 | 4 |    |
|    | G3 | 4 | 5 |    |
| C2 | G2 | 5 | 1 | 39 |
|    | G3 | 4 | 7 |    |
|    | G4 | 3 | 2 |    |
| C3 | G1 | 6 | 3 | 33 |
|    | G2 | 5 | 3 |    |
| C4 | G4 | 3 | 4 | 19 |
|    | G5 | 2 | 1 |    |
|    | G6 | 1 | 5 |    |

## Evaluation results and configuration

The experimental results for the evaluation scenarios of Table 3 are provided in Figs. 6–17. First, we present the PDR of IEEE, GMAC-IEEE and GMAC-IEEE EA in Fig. 6, and of ISA, GMAC-ISA and GMAC-ISA EA in Fig. 7. GMAC-IEEE and GMAC-IEEE EA generate higher values of PDR compared with IEEE with slight superiority of GMAC-IEEE EA. This is interpreted by the geographical distribution awareness of G-MAC based protocols which enable less collision due to dividing them into geographical clusters with less overlap of their coverage compared with one cluster contention in IEEE.

- Packet Delivery Ratio. Both GMAC protocols generate a PDR between 90% and 100%, whereas the PDR of IEEE is as low as 75% in scenarios 3–150 and as low as 30% in scenarios 4–500. Similarly, higher values of PDR for GMAC-ISA and GMAC-ISA EA are obtained compared with basic ISA. ISA and UDG have lower PDR for scenarios 3–400 and 5–400. This is interpreted by the high number of nodes that are distributed in physical clusters where ISA and UDG do not consider clustering them in the MAC scheduling. This causes pressure on the sink. In GMAC with clusters and cluster head, the pressure is mitigated. Moreover, higher PDR is generated from IEEE-based protocols in Fig. 6 compared with the PDR generated from ISA-based protocol in Fig. 7. This is interpreted by the packet lifetime of the latter, which causes less opportunity of sending packets compared with IEEE protocol that does not add lifetime on the packets.

- End-to-end Delay. The second metric that is generated is end-to-end (e2e) delay, which is shown in Fig. 8 for IEEE-based protocols and in Fig. 9 for ISA-based protocol. In all scenarios in Fig. 8, there is a slight difference between GMAC-IEEE's e2e delay and IEEE, indicating the competitive performance between the approaches in terms of the delay. In Fig. 9, the difference in the delay between ISA GMAC-based protocols and ISA is higher that the corresponding results of IEEE, which is interpreted that the packet lifetime shows more influence in ISA experiment in reducing the delay compared with IEEE. In some scenarios, GMAC has lower e2e delay than the original IEEE. For example, the least e2e delay for GMAC-IEEE and GMAC-IEEE EA is for scenario 5–150, that is, 10 s, which is lower than 3 s of the least e2e delay of IEEE. This is interpreted by the capability of GMAC in enabling higher successful transmission with less number of trials compared with IEEE.

- Energy Consumption. The energy consumption is depicted in Figs. 10 and 11 for GMAC-based approaches and the original IEEE and ISA. In Fig. 10, the energy consumption is much lower for GMAC-IEEE EA, and GMAC-IEEE than IEEE. We also observe that IEEE consumes the highest energy for all scenarios of five clusters, namely, 5–200, 5–300, and 5–400 with some differences caused by the random changing in the geographical distribution of the node. However, GMAC-based protocols are less affected by the change in the number of nodes and number of clusters, and their energy consumption level is below 100 W/h for all scenarios. Furthermore, we observe in scenario 5–300 that the energy consumption is 28.1 W/h for GMAC-IEEE EA and GMAC-IEEE but around 578 W/h for IEEE. Similarly, in Fig. 11, the energy consumption for ISA-based protocols, that is, ISA and UDG, are the most consuming for scenarios 5–150, 5–200, and 5–300, indicating the effect of diverse localization area of the nodes and lack of grouping them within clusters on the energy consumption. However, GMAC-ISA is less affected because of its clustering-aware scheduling. Another observation is that GMAC-ISA EA has high energy consumption in some scenarios, such as 3–400, which is interpreted by its behavior in exploiting the residual energy of the nodes for sending that may cause more collision when the nodes reach a state of low energy level in the entire network. Another interpretation factor is the

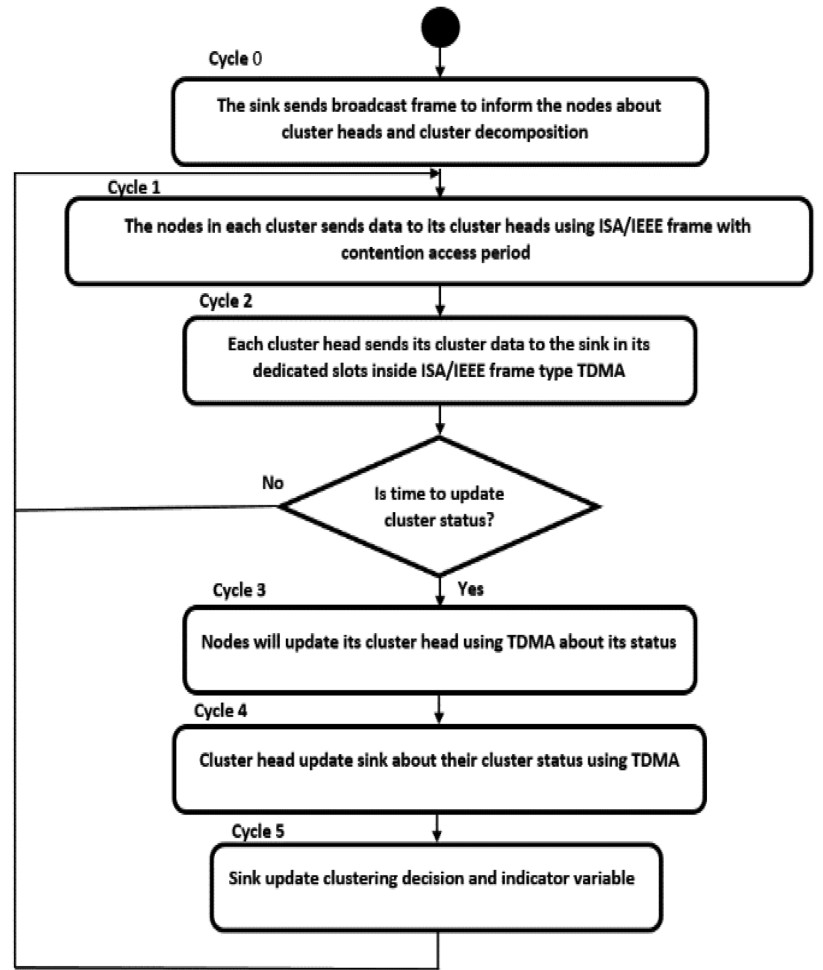

**Figure 4 Activity diagram of the GMAC protocol.**

differences in the geographical positions of the nodes that cause non-linear energy consumption profile. More specifically, the changing in the node locations is another factor that causes variation in the energy consumption and to the number of nodes. Furthermore, the latter factor has non-linear component in the overall energy consumption due to the piecewise nature of the energy consumption formula.

• Network Lifetime. The other metric that is generated is the lifetime, which indicates the experiment time until the first node exits the network. In Fig. 12, GMAC-IEEE and GMAC-IEEE EA experience the same lifetime, which is in general longer than the lifetime of IEEE. This is similar to the behavior of GMAC-ISA and GMAC-ISA EA compared with ISA as in Fig. 13. UDG has a longer lifetime. The pro-organizing aspect of GMAC interpreted the behavior before enabling contention compared with the other protocol that limits the number of collision and prolong the lifetime of the network. The number of retransmissions, which indicates the collision level in the network, is depicted in Figs. 14 and 15. The retransmission numbers for IEEE and ISA

| Algorithm 1 Pseudo code of the GMAC-ISA/IEEE protocol. |
| --- |
| Input |
| Locations of sensors and their initial battery levels |
| Fixed sink |
| Start |
| while (sink is active) |
| 1-the sink node sends broadcast frame to inform about cluster heads and cluster decomposition |
| 2-the node in each cluster sends data to its cluster heads using ISA/IEEE frame with CAP |
| 3-each cluster head sends its cluster data to the sink in its dedicated slots inside ISA/IEEE frame TDMA |
| 4-if (time to update cluster head) |
|     4.1 in each cluster, nodes will update the cluster head about their status using TDMA |
|     4.2 cluster head updates sink about the cluster status using TDMA |
|     4.3 sink updates clustering decision and indicator variables |
| else |
| go to 2 |
| 5-end |
| End |

| Table 2 Simulation parameters that are used for evaluation. | |
| --- | --- |
| Parameter | Value |
| Max Packet Life Time (Max PLT) | 30 s |
| Inter arrival time interval (T) | 5–0.25 s |
| Alert priority levels | 16 |
| Number of priority groups (Ng) | $L_i$ = 1, 2, … 10 |
| Number of packet priority levels | 16 |
| Superframe duration (SD) | 0.25 s |
| Back-off symbol duration | 0.01 s |
| CCA duration | 0.0 0 0128 s |
| ACK packet size | 18 Bytes |
| Packet size | 127 Bytes |
| Data rate | 250 kbps |
| Simulation time | 50 s |
| macMinBE | 3 |
| macMaxBE | 5 |
| Timeslot duration (TD) | 0.01 s |
| Fading type | Nakagami |

are higher than those of GMAC and GMAC EA (IEEE and ISA). This is correlated with the energy consumption of the protocols. As stated earlier, this indicates the role of clustering-aware GMAC in mitigating the collisions in the contention.

**Table 3 Scenarios for internal evaluation of GMAC.**

| Scenario no | Number of cluster-number of nodes |
|---|---|
| 1 | 3–150 |
| 2 | 3–200 |
| 3 | 5–200 |
| 4 | 3–300 |
| 5 | 3–400 |
| 6 | 5–150 |
| 7 | 5–300 |
| 8 | 5–400 |

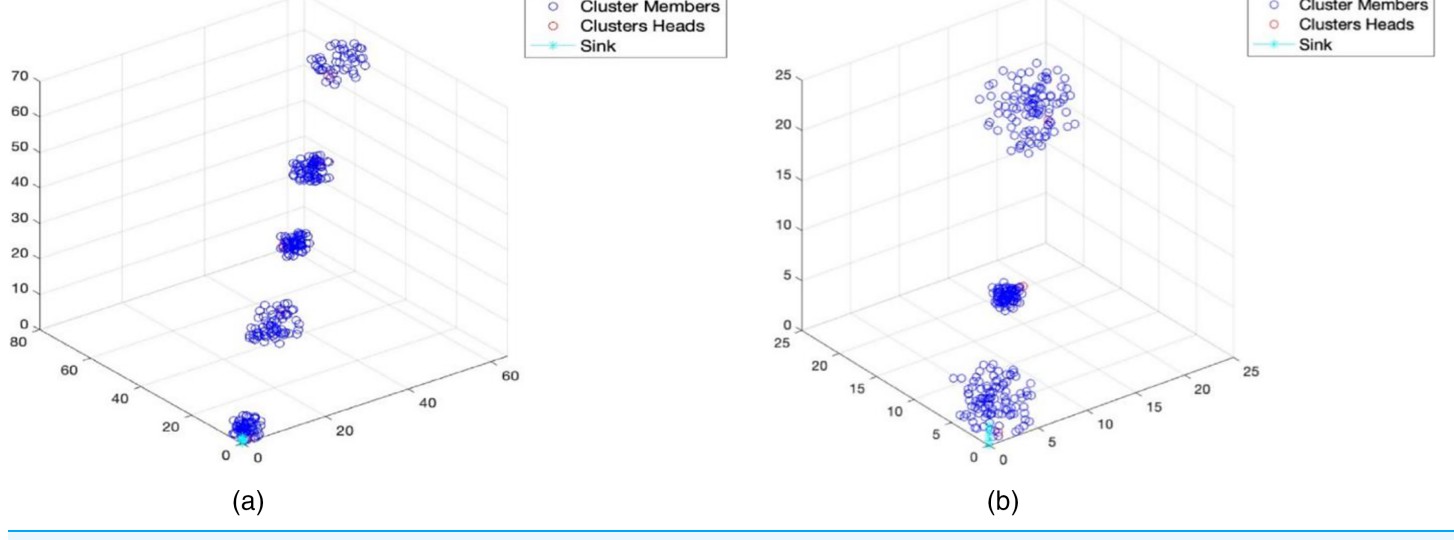

(a)  (b)

**Figure 5 Visualization scenario with number of clusters: 3 and 5; and number of nodes: 300.**

- Throughput. The last metric that is generated is the throughput, which indicates how much of the bandwidth is exploited in producing successful transmission. The results of this metric for IEEE-based and ISA-based protocols are depicted in Figs. 16 and 17, respectively. Both figures reveal that GMAC and GMAC EA have higher throughput than the basic protocols including UDG, regardless of the scenario. The highest achieved throughput by GMAC-IEEE EA and GMAC-IEEE is 539 and 529 kbps, respectively, compared with only 68.4 kbps by IEEE. Similarly, Fig. 17 demonstrates that GMAC-ISA and GMAC-ISA EA have accomplished throughput of 352.7 and 350.2 kbps, compared with 19.8 kbps and 13.8 kbps by ISA and UDG, respectively. Obviously, the high throughput is achieved because of the effective management architecture of nodes access to the medium and its tight relation with the nodes geographical location which is an ignored factor in the original IEEE and ISA.
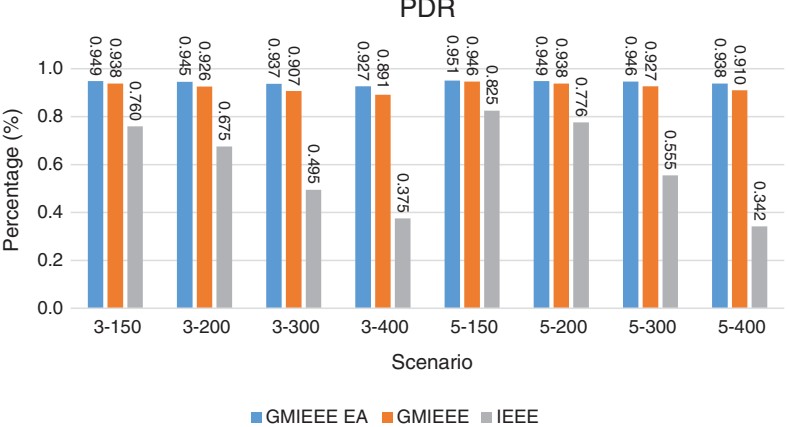

**Figure 6  PDR for GMAC and GMAC EA where the basic protocol is IEEE.**

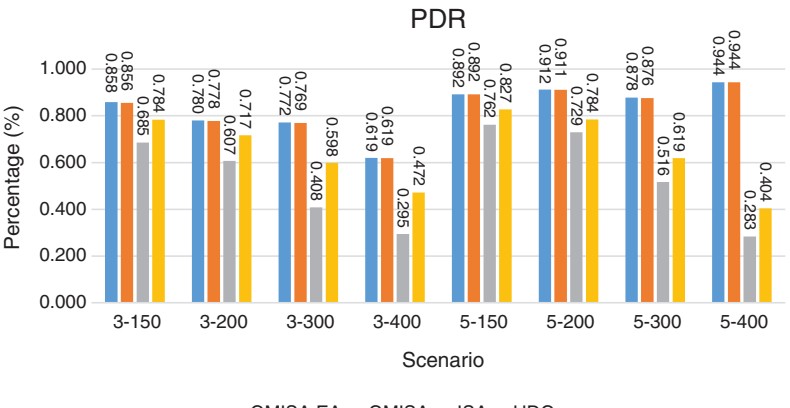

**Figure 7  PDR for GMAC and GMAC EA where the basic protocol is ISA.**

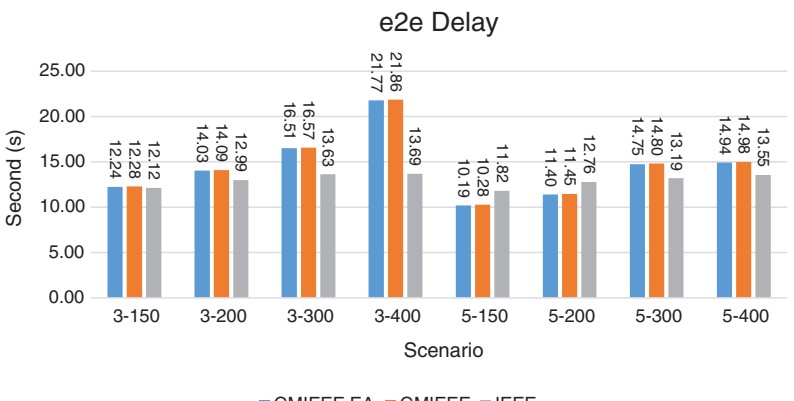

**Figure 8  e2e delay for GMAC and GMAC EA where the basic protocol is IEEE.**

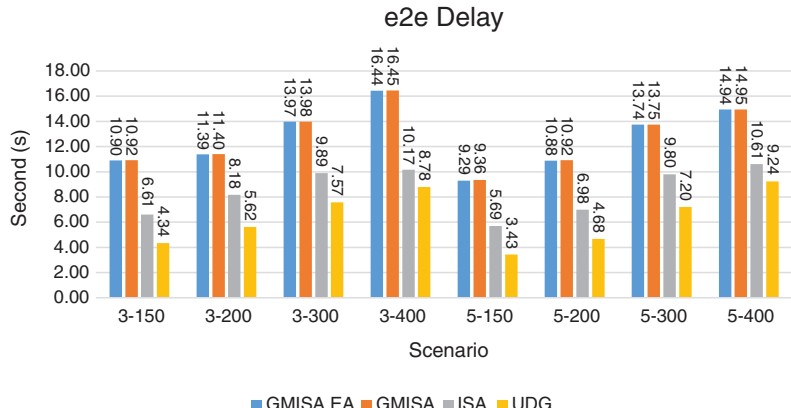

**Figure 9  e2e delay for GMAC and GMAC EA-where the basic protocol is ISA.**

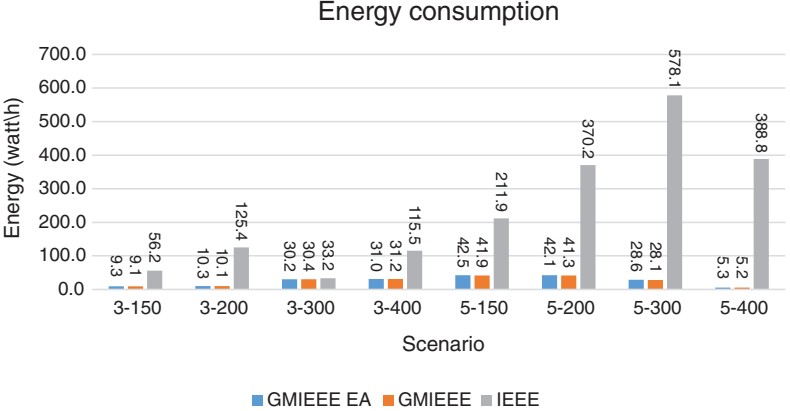

**Figure 10  Energy consumption for GMAC and GMAC EA where the basic protocol is IEEE.**

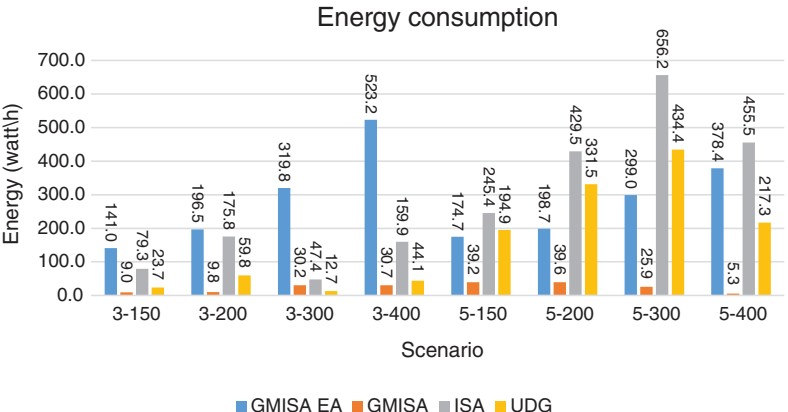

**Figure 11  Energy consumption for GMAC and GMAC EA where the basic protocol is ISA.**

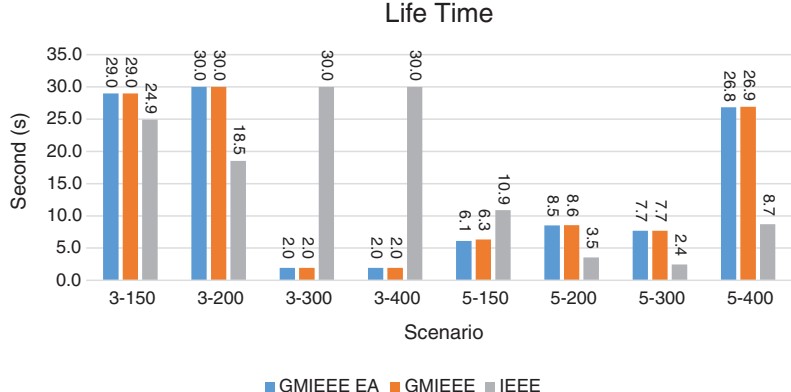

**Figure 12 Lifetime for GMAC and G-MAC-EA where the basic protocol is IEEE.**

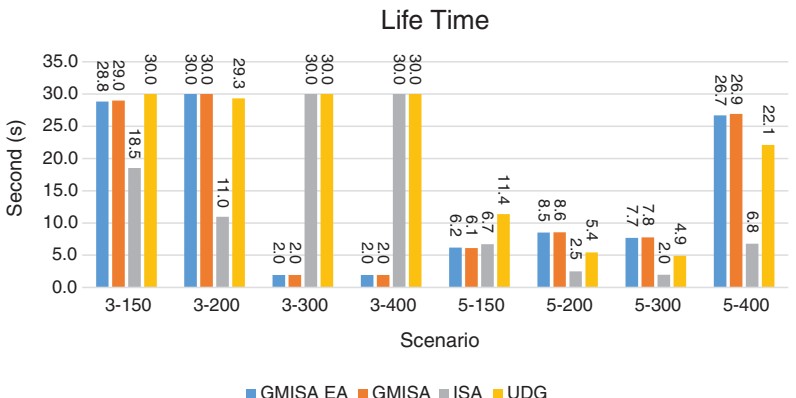

**Figure 13 Lifetime for GMAC and G-MAC-EA where the basic protocol is ISA.**

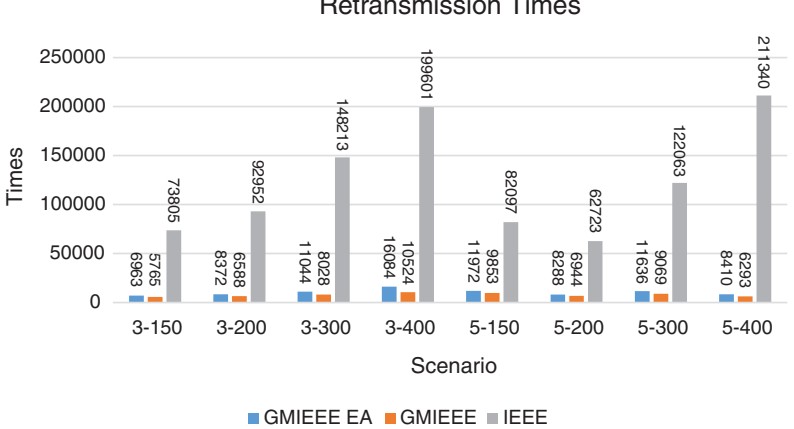

**Figure 14 Retransmission times for GMAC and GMAC EA where the basic protocol is IEEE.**

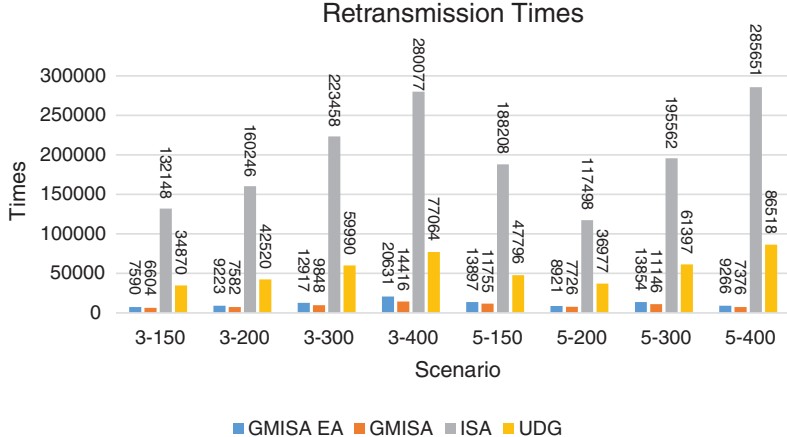

**Figure 15  Retransmission times for GMAC and GMAC EA where the basic protocol is ISA.**

## Throughput

**Figure 16  Throughput for GMAC and GMAC EA where the basic protocol is IEEE.**

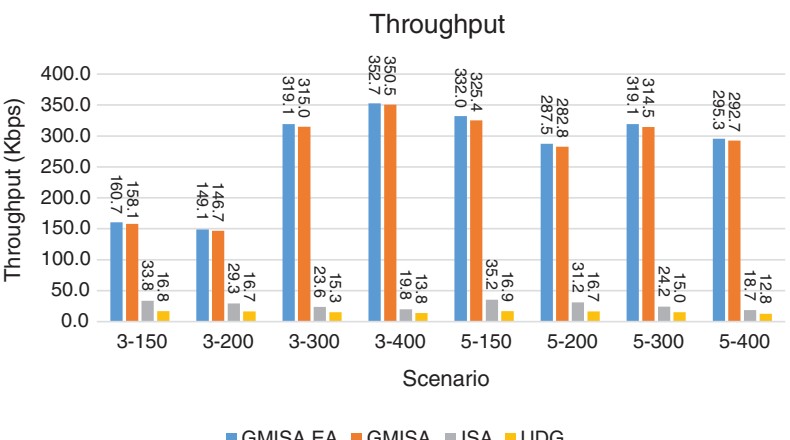

**Figure 17  Throughput for GMAC and GMAC EA where the basic protocol is ISA.**

## CONCLUSIONS

This study proposes a novel protocol for MAC layer in IoT network considering three aspects of operations: low energy level of nodes, high number of nodes, and need of grouping the nodes according to their priorities. This protocol is considered distinct from state-of-the art MAC protocols for IoT, such as IEEE 802.15.4 and ISA 100.11a. In addition, GMAC is suitable to work using SDN technology where a control plane is separated from the data plane and the network is managed by an SDN controller. This is because the protocol requires adaptability to the geographic node distribution by incorporating clustering awareness in the frames and cycles. In addition, GMAC decomposes the network into cycles to enable efficient medium sharing and competition by activating channel access attempts within the cluster that is supposed to be isolated from other clusters in terms of communication collision when a low energy level is used. This is sufficient to transfer packets from nodes to cluster heads while transferring packets to the sink uses dedicated cycles by cluster heads. GMAC is compared with state-of-the-art protocols with respect to various scenarios in terms of number of nodes and priority levels. A clear superiority is observed in PDR, energy consumption, and competitive performance in terms of e2e delay. GMAC protocols generate a PDR higher than 90%, whereas the PDR of benchmark is as low as 75% in some scenarios and 30% in others. In addition, GMAC protocols has lower e2e delay than the least e2e delay of IEEE with a difference of 3 s. Regarding energy consumption, the consumed energy is 28.1 W/h for GMAC-IEEE EA and GMAC-IEEE, which is lower than that of IEEE 802.15.4 (578 W/h) in certain scenarios. The highest achieved throughput by GMAC-IEEE EA and GMAC-IEEE is 539 and 529 kbps, respectively, compared with only 68.4 kbps by IEEE. As limitations for the work, we state that it requests that the nodes are distributed in a wide geographical area where their various clusters have no overlapping in the coverage, so the intra-cluster operations are executed without collisions with other clusters. Second, it assumes that the nodes are stationary, which does not change the clustering decision. However, it enables dynamic reallocation of the cluster head. Future work can develop more energy-saving techniques and incorporate machine learning for channel access. Another future work is to implement hardware for validating the new protocol and evaluate it in real-world scenarios. Developing the work to include channel hopping is an additional future work.

### Funding

This work was supported by the Ministry of Higher Education (MoHE), under Fundamental Research Grant Scheme FRGS/1/2018/TK04/UKM/02/17. The funders had no role in study design, data collection and analysis, decision to publish, or preparation of the manuscript.

## Grant Disclosures

The following grant information was disclosed by the authors:

Ministry of Higher Education (MoHE): FRGS/1/2018/TK04/UKM/02/17.

## Competing Interests

The authors declare that they have no competing interests.

## Author Contributions

- Abdulrahman Sameer Sadeq conceived and designed the experiments, performed the experiments, analyzed the data, performed the computation work, prepared figures and/or tables, authored or reviewed drafts of the paper, and approved the final draft.
- Rosilah Hassan conceived and designed the experiments, performed the experiments, analyzed the data, performed the computation work, prepared figures and/or tables, authored or reviewed drafts of the paper, and approved the final draft.
- Azana Hafizah Mohd Aman conceived and designed the experiments, performed the experiments, analyzed the data, performed the computation work, prepared figures and/or tables, authored or reviewed drafts of the paper, and approved the final draft.
- Hasimi Sallehudin conceived and designed the experiments, analyzed the data, prepared figures and/or tables, authored or reviewed drafts of the paper, and approved the final draft.
- Khalid Allehaibi analyzed the data, prepared figures and/or tables, and approved the final draft.
- Nasser Albogami analyzed the data, prepared figures and/or tables, and approved the final draft.
- Anton Satria Prabuwono performed the computation work, prepared figures and/or tables, and approved the final draft.

## Data Availability

The codes and results and the raw data are available in the Supplemental Files.

## Supplemental Information

Supplemental information for this article can be found online at http://dx.doi.org/10.7717/peerj-cs.733#supplemental-information.

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
