# Peer review of "MAC protocol with grouping awareness GMAC for large scale Internet-of-Things network"

_PeerJ Computer Science, doi:10.7717/peerj-cs.733_

## Round 0.1 · original submission · Major Revisions

Dear Authors,

Kindly consider the comments of all reviewers critically, as the paper requires major modification, and there is room for improvement almost in all areas.

Best Regards
Dr Noor Zaman

·

Basic reporting

The Research Paper is Ok. And novel protocol is being proposed, but the paper needs more technical revisions and is subject for re-review.
Literature review, and the proposed work needs more technical structure information to be elaborated.

Experimental design

Experimental design is Ok. But needs more technical layout.

Validity of the findings

Results are Ok. But analysis part is weak. And needs to be discussed with more detailed description.

Additional comments

The Paper needs the following Strong Revisions and is subject for re-review, and after re-review, the final decision for the paper will be taken:

1. Introduction should be added with more information with regard to the Problem Definition, scope and even with some Background. Add Objectives of the paper at the end of Introduction. Add Organization of the paper.

2. Min 15-25 papers should be cited under literature review, and every paper should be elaborated with what is being observed, what is the novelty aspect and what experimental results are observed. At the end of literature review, highlight what overall technical gaps are observed that led to the design of the proposed methodology.

3. Give a Novel name of the proposed protocol---And add this in Title, Abstract and other sections uniformly in the paper. And under proposed protocol- Give the heading of the proposed protocol--Start with System Model. Then highlight the Architecture- Give Steps, Flowchart and Algorithm of the proposed protocol.

4. Under experimentation- Add Simulation Parameters in Table.
Add assumptions of simulation.
Give the detailed description of every point of testing under seperate head.
Give the Data based table of the values of experimentation.

5. Add some Analysis section and do performance evaluation with more existing proposed protocols/.

6. Add future scope to this paper.

7. Add some more latest references to the paper.

Reviewer 2 ·

Basic reporting

Topic selection is good and basic impression of paper falls positive.

Experimental design

Proposed algorithm must be properly explained in text.

Validity of the findings

Fine

Additional comments

Highlight all assumptions and limitations of your work.
Mention time complexity of entire pipeline.
Mention all figures properly in text.

Reviewer 3 ·

Basic reporting

Professional Proofread and English are required. References and literature review is sufficient.

Experimental design

The rationale, aim, and objectives may need to be clearly defined in the Abstract and other sections.

Validity of the findings

Consistency is missing in the paper. Conclusion, Limitation, and Future scope needs revision for more clarity and enhancing the readability of the paper.

Additional comments

The authors are working on an important area of research. Few observations:
1. The main rationale/objective of the paper needs to be added in the abstract and Introduction.
2. The writing is ambiguous and more clarity is required to enhance the readability of the paper.
3. Consistency is missing in the paper. Conclusion, Limitation, and Future scope needs revision and elaborations for more clarity and enhancing the readability of the paper.
4. Mathematical Equations and paper require thorough proofread.
5. Figure 1 is talking about its application in healthcare, though in the paper authors mentioned having its applications in various domains, then rationale of providing only for healthcare is missing.

---

## Round 0.2 · Minor Revisions

The authors addressed all the comments, and the reviewers feedback is positive on this. The paper is nearly ready for acceptance.

Please pay close attention to your figures and ensure they are clear, with all axes labelled appropriately. Ensure the number of significant figures included is consistent throughout your figures. Axes should be labelled with the variable, with units in brackets (where appropriate). Fig. 15 has overlapping text.

·

Basic reporting

The Revised Paper has presented the Literature review, structure and results in better manner.

Experimental design

Yes, the results are satisfactory.

Validity of the findings

The findings are Ok.

Reviewer 2 ·

Basic reporting

All the changes have been done. Current form of paper is suitable for publication.

Experimental design

All the changes have been done. Current form of paper is suitable for publication.

Validity of the findings

All the changes have been done. Current form of paper is suitable for publication.

Additional comments

Accepted

Reviewer 3 ·

Basic reporting

OK

Experimental design

OK

Validity of the findings

OK

Additional comments

Suggested changes are incorporated. We may proceed.

---

## Round 0.3 · accepted · Accept

The authors address all the concerns and comments carefully. The paper stands for acceptance.